# Restraint upon Embryonic Metatarsal *Ex Vivo* Growth by Hydrogel Reveals Interaction between Quasi-Static Load and the mTOR Pathway

**DOI:** 10.3390/ijms222413220

**Published:** 2021-12-08

**Authors:** Soraia Caetano-Silva, Bigboy H. Simbi, Neil Marr, Andrew Hibbert, Steve P. Allen, Andrew A. Pitsillides

**Affiliations:** Comparative Biomedical Sciences Department, Royal Veterinary College, London NW1 0TU, UK; bsimbi@rvc.ac.uk (B.H.S.); nmarr@rvc.ac.uk (N.M.); ahibbert@rvc.ac.uk (A.H.); sallen@rvc.ac.uk (S.P.A.); apitsillides@rvc.ac.uk (A.A.P.)

**Keywords:** endochondral ossification, mTOR, NF-ĸB, quasi-static, loading, hydrogel

## Abstract

Mechanical cues play a vital role in limb skeletal development, yet their influence and underpinning mechanisms in the regulation of endochondral ossification (EO) processes are incompletely defined. Furthermore, interactions between endochondral growth and mechanics and the mTOR/NF-ĸB pathways are yet to be explored. An appreciation of how mechanical cues regulate EO would also clearly be beneficial in the context of fracture healing and bone diseases, where these processes are recapitulated. The study herein addresses the hypothesis that the mTOR/NF-ĸB pathways interact with mechanics to control endochondral growth. To test this, murine embryonic metatarsals were incubated *ex vivo* in a hydrogel, allowing for the effects of quasi-static loading on longitudinal growth to be assessed. The results showed significant restriction of metatarsal growth under quasi-static loading during a 14-day period and concentration-dependent sensitivity to hydrogel-related restriction. This study also showed that hydrogel-treated metatarsals retain their viability and do not present with increased apoptosis. Metatarsals exhibited reversal of the growth-restriction when co-incubated with mTOR compounds, whilst it was found that these compounds showed no effects under basal culture conditions. Transcriptional changes linked to endochondral growth were assessed and downregulation of *Col2* and *Acan* was observed in hydrogel-treated metatarsi at day 7. Furthermore, cell cycle analyses confirmed the presence of chondrocytes exhibiting S-G2/M arrest. These data indicate that quasi-static load provokes chondrocyte cell cycle arrest, which is partly overcome by mTOR, with a less marked interaction for NF-ĸB regulators.

## 1. Introduction

Endochondral growth, which occurs in all skeletal limb elements, involves early mesenchymal cell condensation, as well as sequential proliferation, differentiation into cartilage-forming chondrocytes with linked extracellular matrix (ECM) production, and later, hypertrophy. In this sequence, the cartilage ECM is ultimately calcified and invaded by blood vessels and is thus primed for subsequent ossification. In addition, these endochondral processes are critical at the expanding epiphyses, where long bone lengthening continues until maturity, and are also recapitulated in pathology during bone fracture healing and osteoarthritis. Despite these vital roles, the mechanical factors controlling endochondral growth are incompletely defined, and the signalling pathways via which the emergent mechanics exert local control are only partly resolved.

Endochondral growth chondrocytes which express SRY-box transcription factor 9 (SOX9), bone morphogenic proteins (BMPs) and fibroblast growth factor (FGF) [1,2,3] are organised with respect to their proximity to the ossifying regions. Resting chondrocytes are more distant, constituting a reservoir of skeletal stem cells expressed under parathyroid hormone-related protein (PTHrP) [4]. These undergo sequential proliferation, differentiation, and hypertrophy, with those adjacent to the resting zone dividing intensively, rearranging into columns under PTHrP control, which delays further differentiation. They then enter a post-mitotic phase, synthesising large amounts of aggrecan (ACAN) and collagen-II (COL-II) and expressing Indian hedgehog (IHH) that, in turn, promotes resting zone PTHrP. Chondrocytes near ossifying zones become hypertrophic, at which point they express collagen-X (COL-X), IHH, runt-related transcription factor 2 (RUNX2), alkaline phosphatase (ALPL), matrix metallopeptidase 13 (MMP13) and vascular endothelial growth factor (VEGF) [4,5,6,7,8].

Specific connections between endochondral growth and local mechanical stimuli are emerging. Mechanical stimuli have, for example, been shown to regulate cartilage growth [9,10,11], and animal models in which movement is restricted show impaired growth [12,13,14,15,16]. Despite these connections, the candidate regulators through which mechanical factors interact to control endochondral growth are ill-defined. Cultured mouse embryo metatarsi are an alternative *ex vivo* model that avoids systemic effects and the need to contend with the placenta/embryo barrier assay [17,18]. Using this embryo metatarsal model, it has been shown that known *in vivo* skeletal growth promoters, namely growth hormone (GH) and insulin-like growth factor 1 (IGF-I), increase longitudinal expansion [19]. In contrast, this model has disclosed that a PI3K/MEK-ERK pathway inhibitor co-treatment (LY294002/PD98059) severely restricts growth [20], demonstrating that the model is both amenable to exogenous pharmacological intervention and that its use may allow interactions between mechanical stimuli and specific pathways that can be selectively examined. 

Inflammation-related pathways can be stimulated by mechanical cues, such as nuclear factor kappa-light-chain-enhancer of activated B cells (NF-ĸB), a transcription factor which is ubiquitously expressed and central to such responses. NF-ĸB signalling is connected to Akt, which can serve to activate the mTOR (mTORC1) pathway; this has been shown to contribute to endochondral growth [21,22,23,24].

Transcriptional profiling reveals downregulation of mTOR components upon acquisition of mechanical sensitivity of endochondral growth in embryo limbs [25]. These data imply that mTOR activity in individual endochondral growth zones matches the extent to which expansion is controlled mechanically, rather than solely by intrinsic cues. Studies indicating that genetic selection for intrinsically high proliferation and growth rates occurs at the expense of mechanosensitivity [26] support this view. Potential mTOR regulation of cell cycle kinetics via mechanical cues has not hitherto been examined. 

It has also been shown that mechanical control of endochondral growth can, however, be regulated by mTOR, which is necessary for the mechanical regulation of embryonic cartilage function [27,28,29,30]. NF-ĸB has also been reported to control normal endochondral growth in limb elements through chondrocyte proliferation and differentiation [31,32]. Here, we explore whether *ex vivo* imposition of quasi-static load discloses interactions between mechanical stimuli and nutrient-sensing mTOR and/or pro-inflammatory NF-ĸB pathway signalling in the control of endochondral growth. We examine if mTOR/NF-ĸB modulators modify endochondral growth and establish a novel hydrogel-based system for the quasi-static loading of metatarsals where mechanical–mTOR/ NF-ĸB pathway interactions are explored. Assessment of transcriptional markers of endochondral growth and cell cycle kinetics define novel resident chondrocyte behaviours regulated by quasi-static mechanical load and selective pathway interactions.

## 2. Results

### 2.1. Mouse Metatarsal Growth Is Relatively Insensitive to mTOR and NF-ĸB Pathway Regulators

Previous studies have used E15 stage mouse metatarsals to examine skeletal dynamics [17,33,34,35,36]. To establish if metatarsals from a later stage can simplify isolation yet preserve *ex vivo* growth behaviours, metatarsi from stage E17 embryos were cultured and found to show significant ~40% length increases (over 14 days), equating to ~3%/day (~45 µm/day; ~1500 µm at the outset). Variation in growth kinetics was seen but a relatively small overall deviation (6% coefficient) was evident in individual metatarsi after 14 days. Sensitivity to exogenous factors in E17 metatarsi was confirmed by significant growth inhibition in response to combined LY294002/PD98059 treatment (Figure 1a). Further validation by comparison to post-natal (P0, P5 and P10) *in vivo* growth showed that metatarsi grew less *ex vivo* but nonetheless significantly increased their total, mineralization and cartilage zone lengths (Figure 1b); they showed >two-fold increased total lengthening from P0–P10—which was negligible at early stages (P0–P5), but also showed >3-fold mineralisation zone expansion from P5–P10 and, contrastingly, cartilage zone lengthening at only P0–P5 (Figure 1b). Cartilage zone expansion was the main contributor (~50% *in vivo**)* during the 14 days in culture.

Intra-assay screening was used to first identify the concentrations of mTOR/NF-ĸB regulators that may modify longitudinal murine metatarsal expansion (Table 1); total metatarsal length at day 14 of culture was measured and linear mixed effect statistical analysis was applied. Cumulative data (*n* = 8 experiments) were pooled and the impact of mTOR and NF-ĸB regulators on total, mineralised and cartilaginous metatarsal zone lengths analysed to reveal that leucine, rapamycin, betulinic acid and SC-514 exhibited no change (vs. controls) in total or cartilage zone length (Figure 1c). Intriguingly, a small yet significant increase in mineralisation zone length in metatarsi treated with the NF-ĸB inhibitor SC-514 was observed (Figure 1c). These data were strengthened by significant increases in total, mineralised and cartilage zone lengths between days 0–14.

### 2.2. Interaction between Quasi-Static Loading and mTOR Modulates Endochondral Growth

Initial long-term growth monitoring identified VitroGel3D™ as a suitable hydrogel through which quasi-static load could be delivered. This was apparent in the resistance placed upon the longitudinal metatarsal expansion that would otherwise have occurred under *ex vivo* conditions. This hydrogel was the most practical way of testing mechanics in these embryonic tissues. Metatarsi maintained in a 1:1 hydrogel solution showed an almost total arrest of longitudinal expansion during the 14 days (control 2.32 ± 0.36 and hydrogel 1.62 ± 0.37, mean ± SD; Figure 1d). Based on this restriction upon expansion, and assuming linear growth, these hydrogel conditions represent the imposition of quasi-static compressive strains of ~30.4% over 14 days, which equates to strain rates of ~2%/day or ~900 με/hz. To interrogate the effects of hydrogel concentration, metatarsi were maintained in varying dilutions (1:1–1:300). This resulted in a full growth arrest in dilutions up to 1:20, with a sharp decline in growth arrest at the 1:50 dilution, wherein metatarsi gained lengths similarly to controls; hereafter, data from the 1:1 dilution are reported, unless otherwise stated (Figure 1e).

To test if the quasi-static load interacts with mTOR and NF-ĸB pathways to regulate endochondral growth, the influence of leucine, rapamycin, betulinic and SC-514 was re-explored in the presence of hydrogel-mediated growth restraint. This revealed significant interactions with mTOR, as addition of rapamycin not only produced greater rates of total length-wise growth but also greater lengthening of the mineralisation zone than in hydrogel alone (Figure 1c). Similar data were intriguingly observed when metatarsi were supplemented with leucine, which caused significantly greater total lengthening (than hydrogel alone), but unlike rapamycin, instead generated significant increases in cartilage zone length only (Figure 1c). Changes in total, mineralisation and cartilage lengths were consistently higher in metatarsi cultured in a combination of either leucine or rapamycin and hydrogel (vs. hydrogel alone; Figure 1c). In contrast, the NF-ĸB modulators SC-514 and betulinic acid did not modify any parameter in the presence of hydrogel-induced quasi-static restraint. These data indicate significant mTOR pathway interactions with quasi-static load in the control of EO.

Arrests in metatarsal growth by hydrogel may involve changes in cell proliferation, viability, or survival. To explore this further, PCNA labelling and whole-mount 3D confocal microscopy was performed to pinpoint chondrocytes in the S-phase [37]. This (PCNA, red; nuclear DAPI, blue; co-labelling white) showed little if any reduction in PCNA labelling in hydrogel-treated samples (vs. control; Figure 2a). Quantification of total volume (µm^3^) and mean fluorescence intensity showed that the mean PCNA and DAPI labelling intensity was greater in hydrogel-treated metatarsi, indicating a greater density or closer packing of S-phase chondrocytes under quasi-static load conditions (Figure 2a). Assays employed to determine resident chondrocyte viability demonstrated that chondrocytes in control and hydrogel-treated metatarsi were labelled predominantly ‘live’, with little, if any, ‘dead’ cell labelling (Figure 2b). Assessment of FITC/Texas Red cell labelling intensity (RFU/µm^2^) showed that metatarsi in hydrogel (14 days) did not exhibit changes in viability nor greater levels of cell death than controls (Figure 2b). These data suggest that hydrogel-related growth arrest does not involve changes in resident cell proliferation, viability, or survival.

### 2.3. Transcriptional Analyses Support the Interplay between Quasi-Static Mechanical Load and mTOR Pathway Signalling in the Control of Endochondral Growth

We also assessed mRNA expression levels linked to endochondral growth by multiplex qRT-PCR in metatarsi cultured in hydrogel, with genes arbitrarily organised based on their relatedness to the matrix content (*Col2*, *Acan*, *Col10*, *Alpl*, *Mmp13*, *Tbsp4*) or their established regulatory roles (*Pthrp*, *Sox9*, *Ihh*, *Runx2*, *Vegf*, *Akt* and *Il6*). Principal component analysis demonstrated the presence of three major clusters, comprising: (i) *Acan*, *Alpl*, *Col10* and *Mmp13*; (ii) *Runx2*, *Col2*, *Sox9* and *Akt;* and (iii) *Ihh* and *Pthrp,* respectively, with three outliers (*Vegf*, *Il6*, *Tbsp4*; Figure 3a). *Col2* and *Acan* alone were found to confer 72% of total variance, and exploration of the scale and direction of the interactions using a correlation matrix reinforces the three major clusters. *Col2* and *Acan* showed significant decreases in hydrogel at day 7 (Figure 3b). *Mmp13* levels were significantly lower in hydrogel + rapamycin treatment than hydrogel-treated metatarsi at day 7, and yet were raised in hydrogel + leucine compared to hydrogel alone at day 14 (Figure 3c). *Il6* levels were significantly lower in hydrogel + leucine than in hydrogel alone (Figure 3d), which aligns with the raised IL-6 protein levels measured in media conditioned by metatarsi in hydrogel (vs. levels with mTOR modulators added; Figure 3d). These data suggest that quasi-static load reveals the mechanical interplay between mTOR pathway signalling components in the regulation of endochondral metatarsal growth.

### 2.4. Quasi-Static Loading Induces Chondrocyte S to G2/M Cell Cycle Arrest

Changes in cell cycle kinetics were examined at 5, 7 and 14 days under control and hydrogel conditions using propidium iodide labelling/flow cytometry. Gating was used to define four putative chondrocyte populations: pre-hypertrophic (P2); proliferative (P3), hypertrophic (P4) and all these populations combined (designated P1). Analysis of P1 showed (Figure 4a) that the percentage of cells in G0/G1 was higher at day 5 and steadily declined at day 7 and 14 in both controls and hydrogel (Table 2); this shift indicates a progression to a raised proportion of cells in G2/M. The percentage of total cells (P1) in S phase was also consistently greater in hydrogel, indicating a prolongation of the S phase. P2 displayed no cells in G0/G1, and a vast proportion in the G2/M phase; the percentage of cells in each of these phases was similar in controls and hydrogel, with fewer cells in the S phase with prolonged culture—particularly between days 5 and 7 (Table 2).

In contrast, P3 contained no cells in the G2/M phase, and consistently more in the S phase in metatarsals maintained in hydrogel (vs. control) at all time-points (Table 2). P4 comprised an almost exclusive subdivision to G2/M and few S phase cells, and no G0/G1 phase was identified. Again, the most marked hydrogel-induced shift was observed on day 5, when marked increases in S phase cells were evident (Table 2). Pooling of P2–P4, as expected, agreed closely with data from P1 (Figure 4b). These data show that the hydrogel induced an expansion of cells in the S phase, most markedly on day 5, which was most pronounced in the smallest chondrocytes, in which these increases seem to be at the expense of time spent in G0/G1. 

## 3. Discussion

Our findings show: (i) that E17 metatarsi preserved a scope for *ex vivo* growth equivalent to 50% of the *in vivo* rate and showed sensitivity to known endochondral growth inhibitors (LY294002/PD98059); (ii) that neither mTOR nor NF-ĸB regulators affected growth under basal conditions; (iii) that quasi-static load via hydrogel exerted growth arrest, which was reversed by mTOR but not by NF-ĸB modulators; and, finally, (iv) that hydrogel-related growth restraint was linked to an S-G2/M cell cycle arrest, itself linked to ECM (*Col2*, *Acan*, *Col10*) transcriptional changes—with those linked to hypertrophy (*Col10*, *Mmp13)* also exhibiting a corresponding reversal by mTOR modulators. We conclude that an applied mechanical cue discloses interactions with the nutrient-sensing mTOR pathway to control endochondral growth by regulating cell cycle progression and hypertrophy of resident chondrocytes.

Our knowledge of the mechanical regulation of endochondral growth remains somewhat sparse and is abridged by the Heuter–Volkmann law: static compression slows, and static tensional forces accelerate longitudinal growth. Studies using load protocols in rat vertebrae and ulnas and porcine mesenchymal stem cells have recently reinforced this view by showing that cyclic tensile strain promotes EO [10], and that both axial loading [38] and static and dynamic compression, likewise, suppress longitudinal growth [39]. The quasi-static loading system we have used regulates endochondral expansion *ex vivo* in a tuneable system for controlled growth regulation. This gives new awareness of and allows questions concerning the interactions between pharmacological interventions and mechanical cues to be tested readily *ex vivo.* This has allowed us to disclose a novel interaction between the nutrient-sensing mTOR pathway and mechanics in the regulation of endochondral growth. Our findings additionally indicate that such an interaction is less extensive via the pro-inflammatory NF-ĸB signalling pathway and provides insights into the mechanisms by which this mechanical crosstalk is achieved in resident cartilage chondrocytes.

Culturing of metatarsi under basal conditions has revealed that activators/inhibitors of mTOR signalling do not to modify any parameters of growth; only SC-514-mediated NF-ĸB modulation resulted in increased mineralisation zone length. Previous studies have shown that SC-514 does not reduce hypertrophy in human osteoarthritic chondrocytes [40], and that the NF-ĸB signalling pathway can control chondrocyte differentiation [41]. Furthermore, it has been reported that differentiated osteoblasts generate increased trabecular bone mass and higher bone mineral density in circumstances where the IKK gene was likewise inhibited [42]. This has been extended by *ex vivo* studies in which SC-514 increased mineralisation by attenuating osteoclast function but also by influencing osteoblastogenesis [43]. These data suggest that basal NF-ĸB activities restrict mineralisation and that their inhibition by SC-514 promotes accelerated levels of mineralisation.

Activators/inhibitors of mTOR signalling did not modify growth under basal conditions. Rapamycin is the best studied of the mTOR inhibitors and leucine is a well-established, natural mTORC1 mTOR complex activator. The lack of any significant effects of rapamycin on longitudinal growth *ex vivo* may also have been due to the concentration used. Rapamycin differentially targets mTORC1 and mTORC2 in a concentration-dependent manner. Thus, 100 nM rapamycin may be low enough to have not inhibited either mTORC1/2, and this is known to allow for compensatory mTOR pathway activation. It has been found that leucine targets mTORC1 to activate cell synthesis and proliferation to accelerate muscle growth [44,45]. There are no studies reporting the effects of leucine on embryonic endochondral growth. It is possible that leucine does not promote growth in metatarsals due to constitutive mTOR activation under *ex vivo* conditions, where nutrient availability in the media is not limiting. In contrast to mTOR, NF-ĸB is an inflammatory pathway regulated by exogenous betulinic acid, which has reported inhibitory or stimulatory activity that depends on cell type and concentration [46,47,48]. Despite this, no effects on endochondral growth were seen with addition of betulinic acid to metatarsal cultures. Other studies using bone cells *in vitro* have found that 10–15 µM and 10 mM betulinic acid induce bone formation by enhancing BMP2 [49], and inhibit osteoclastogenesis via RANKL [50], respectively. Herein, betulinic acid was used at 2.5 µM, which might be too low to exert any effects on endochondral growth in the context of an *ex vivo* organ culture system, which likely shows different requirements to cultured cells *in vitro*.

Our data reveal a very dramatic mechanically-engendered inhibition of longitudinal growth in hydrogel, suggesting that prolonged static compressive loads may evoke an irreversible blockade of EO. Studies in chick embryos have shown that the effects of ‘rigid’ paralysis at early embryo stages, which would likewise engender quasi-static load, could be partially overcome in the spine, but not in developing joints, by restoring movement [16]. This suggests that the dynamic loads associated with movement may reverse some of the effects of quasi-static loading. The effect of quasi-static load imposed by hydrogel may also be reversed if appropriate dynamic stimuli were restored to cultured metatarsi, and perhaps the differences in longitudinal expansion observed may be attributable to the lack of dynamic loading when the metatarsals are maintained in isolation. 

Dilution of hydrogel showed a potential for concentration-dependent growth inhibition. Recent studies have shown the importance of hydrogel stiffness in chondrogenic differentiation and growth, highlighting that the composition and loading regimes are crucial [51,52]. It has been shown that lower stiffnesses, along with higher mesh sizes, allow for larger molecules to diffuse and glucose and oxygen rates to be maintained [53,54]. The fact that 1:20 vs. 1:1 dilution exert similar restraints on longitudinal expansion suggests that this rescue is more likely due to mechanical interactions with the mTOR and NF-ĸB pathways than to diffusion. The results obtained for the 1:20 hydrogel dilution clearly indicate that the hydrogel density is an important factor and that the ‘tunability’ of the quasi-static loading system is a vital asset. Neither betulinic acid nor SC-514 exerted any modification of growth at the 1:1 hydrogel dilution. Dynamic load stimuli are known to activate or suppress NF-ĸB signalling in chondrocytes *in vitro* depending on their magnitude [55]. The quasi-static loading for 14 days used here may thus represent a relatively mild loading regime with regard to activation of NF-ĸB. Although betulinic acid and SC-514 had no effect, mTOR regulators, leucine and rapamycin did; partial reversal of the hydrogel-related growth restraint by these pharmacological mTOR pathway modulators suggests that there is a requirement that the system is perturbed—here, slowed by hydrogel—in order that mTOR’s role in maintaining the *status quo* can be made apparent.

This was also true in the quantification of early and late transcriptional EO-marker mRNA levels. The differences in mRNA levels in metatarsals maintained in hydrogel (vs. control) were more evident at day 7, targeting resting chondrocytes (*Col2* and *Acan*). In all cases, except for *Mmp13,* a downregulation of mRNA levels was seen in hydrogel-treated metatarsi on day 7 and no differences were seen at day 14. Furthermore, addition of leucine at day 14 promoted upregulation of *Col10* and downregulation of *Il6* mRNA levels. Together, these results suggest three key changes in mRNA levels: *Col2, Acan* and *Col10*, with all three suppressed by hydrogel and exhibiting scope for reversal back to control levels of expression upon mTOR modulator intervention. 

The upregulation of the NF-ĸB marker IL-6 in culture supernatants in the presence of hydrogel may be interpreted as evidence for a stress-induced response due to quasi-static load. This would likely be via chronic NF-ĸB pathway activation. Intriguingly, these hydrogel-induced levels of IL-6 release were blocked by the addition of either leucine or rapamycin, which strongly supports the view that crosstalk between mTOR/NF-ĸB exists in the metatarsal only under quasi-static load conditions.

This raises questions about the mechanisms that conserve the long-term viability of resident chondrocytes when growth is significantly limited. Analysis of cell cycle kinetics under quasi-static load conditions exposed an S to G2/M phase arrest, which was more evident in the P3 (proliferative) cell population. Cells in metatarsals within the hydrogel are subjected to what amounts to a continuous compression, which may be translated into a signal instructing how much expansion cells should undergo. To promote longitudinal growth, chondrocytes must proliferate and the data from our cell cycle analyses indicate that this proliferation was halted at S phase by the stiffness of the hydrogel. It is tempting to speculate that the level of proliferation under these quasi-static load conditions maintains sufficient chondrocyte renewal rates and viability, but that less than this would facilitate a significant increase in length. The cell cycle can also be arrested due to DNA damage. If prolonged quasi-static loading induces DNA damage/stress, then arrest at cell cycle checkpoints will ensure a halt upon division. Several checkpoints are enforced during the cell cycle, and one of them is during S phase: the DNA damage checkpoint. This checkpoint functions to coordinate DNA replication and repair and cell cycle progression [56]. Thus, chondrocytes in these quasi-statically loaded metatarsi appear not to successfully complete this S phase checkpoint, resulting in a delay in chondrocyte proliferation until DNA damage is successfully repaired and conditions to facilitate cell cycle progression are again met. It remains to be seen whether these mechanisms underpin the rescue of growth achieved by mTOR modulators at 1:20 hydrogel dilutions. It is also apparent that the tunability of the system and the apparent reversibility of the effects of hydrogel may be exploited to more fully address these mechanisms.

## 4. Materials and Methods

### 4.1. Mice

All animal procedures were performed in accordance with the Home Office guidelines in the UK, the Animals (Scientific Procedures) Act 1986, under PPLs 70/7859, P253385BB2, and PIL I5FB864C0, and were approved by the Royal Veterinary College’s ethics committee. Endochondral ossification was studied *in vitro* using an established model [17]. This involved maintenance of a colony of C57BL/6J mice in the Biological Services Unit, Royal Veterinary College, housed with regulated humidity and a 12 h light and dark cycle under an appropriate breeding programme in a conventional facility, and embryos collected using an established protocol from pregnant mice at day E17 [17]. 

### 4.2. Mouse Metatarsal Organ Culture Model

Briefly, pregnant mice were euthanized by cervical dislocation and embryos were harvested immediately and euthanized by decapitation. The embryos were immediately placed in 50 mL tubes containing sterile PBS (Gibco, Waltham, MA, USA). Once all embryos had been collected, they were transferred to Petri dishes and both hind limbs were separated from the body using scissors. Once hind limbs were collected, they were transferred in sterile PBS to be viewed under a dissection microscope, and with the legs held in place using forceps, the skin was removed from the entire limb and the phalanges removed by pinching and pulling until the metatarsi were exposed using fine forceps (F.S.T. Dumont #4 and #5, Heidelberg, Germany). Second, the third and fourth metatarsi (numbers 2–4, middle) were collected, leaving the perichondrium intact but removing any extraneous soft tissues, and transferred into a new Petri dish containing 1X PBS, and the remaining bones of the limb discarded.

After collecting all the metatarsi from all of the pups from each pregnant mouse (~7 pups), each metatarsus was moved using small forceps to an individual well in a 24-well plate containing α-MEM culture media with or without supplementation; media was changed every other day and cultures maintained for 14 days in medium containing 0.2% BSA (Sigma-Aldrich, St. Louis, MO, USA), 5 µg/mL L-ascorbic acid phosphate (Wako, Osaka, Japan), 0.05 mg/mL gentamicin (Gibco, Waltham, MA, USA) and 1.25 µg/mL amphotericin B (Gibco, Waltham, MA, USA) [17].

### 4.3. Pharmacological Manipulation of Murine Metatarsi

To study the effects of pharmacological manipulation of specific chosen pathways, metatarsals within individual wells were cultured in basal medium supplemented with either: (i) 100 nM rapamycin (MedChem Express, Sollentuna, Sweden), (ii) 2.5 µM betulinic acid (Abcam, Cambridge, UK), (iii) 20 µM SC-514 (Abcam, Cambridge, UK) or (iv) 10 mM leucine (Sigma-Aldrich, St. Louis, MO, USA) [8], or (v) 10 µM LY294002 (LY) and PD98059 (PD; Abcam, Cambridge, UK), after stable *ex vivo* cultures were established. Medium was replenished every other day. NF-ĸB is associated with its inhibitor, IĸBα, when inactive; its activation involves phosphorylation of the IĸB kinase (IKK) complex to release NF-ĸB, leading to nuclear translocation and modified transcription [57]. IKK can be modulated by SC-514, which selectively inhibits NF-ĸB phosphorylation to block activation; whilst it may be stimulated by betulinic acid, which promotes NF-ĸB-mediated transcriptional activity [46,58,59,60,61,62]. mTOR, which is comprised of mTORC1 and mTORC2 complexes, coordinates growth and cell division in response to nutritional and growth factor status. mTORC1 contains the regulatory protein Raptor sensitive to rapamycin and is a key growth factor as well as a mediator of insulin. In contrast, mTORC2 contains the rapamycin-insensitive protein Rictor, and instead activates NF-ĸB via Akt. mTORC1 is thus inhibited by rapamycin and is stimulated by exogenous leucine, which acetylates Raptor via acetyl-coenzyme A [44]. These pathways are thus subject to exogenous regulation and show scope for extensive crosstalk [63]; NF-ĸB can function downstream of Akt (inhibiting mTORC1 suppresses NF-ĸB) but also serve as an upstream regulator of mTOR (IKK activates mTORC1/2; [64].

### 4.4. Mechanical Manipulation of Murine Metatarsi

Additional studies were designed to explore the effects of imposing quasi-static load during growth. This involved the incubation of embryonic mouse metatarsals in both the absence or presence of VitroGel^TM^ 3D (The Well Bioscience Inc, North Brunswick, NJ, USA). At the 14-day endpoint, metatarsi were placed in RNA later^TM^ solution (Invitrogen, Waltham, MA, USA) for subsequent RNA extraction and analysis.

### 4.5. Quantification of Change in Element Length

Photographs of individual metatarsi were taken using a Leica camera adapter on a Leica MZ apo stereo zoom microscope twice each week, to allow for the measurement of the bones’ length with ImageJ software on specific days during *in vitro* culture. All sessions also involved identical image capture of a graduated ruler positioned in the sample plane for calibration. For measurement, each photo was opened and calibrated in ImageJ 1.48v (National Institute of Health, Bethesda, MD, USA) and a straight line used to measure total metatarsal length from distal to proximal extremities. Measurements were also recorded for the length of the mineralizing zone, and subtraction of this mineralised zone length from the total length provided a measure of the cartilage length to yield measurements of the total, mineralised and cartilage zone lengths.

### 4.6. Sample Size and Individual Experiments in Metatarsi Length

Analysis was performed both within individual experiments and in pooled experiments; full information regarding both the total number of metatarsi and number of experimental replicates is provided within each figure legend. Individual experiments were mostly used to provide either confirmatory data, or pilot data regarding the specific effects of a compound or the exploration of singular timepoints. Pooled data from numerous metatarsi from across several replicate experiments with high sample sizes were used to explore the overall effect; appropriate statistical approaches were employed for each.

### 4.7. Live/Dead Assay

To investigate the viability of the cells in the cultured metatarsi, a Live/Dead assay (Abcam, Cambridge, UK) was carried out following the manufacturer’s instructions. Accordingly, metatarsi collected immediately after dissection and the establishment of cultures (day 0) and after 14 days of maintenance under standard culture conditions, were stained with 5× Live and Dead Cell stain in DMSO in 300 µL of 1× sterile PBS, and incubated for 10 min at RT. After the incubation, samples were imaged on a Leica DM4000 upright fluorescence microscope (Leica Biosystems, Nussloch, Germany) with a Zeiss mRM camera using Zeiss ZEN 3.1 blue edition software (Carl Zeiss Microscopy GmbH, Munich, Germany) with L5 (green fluorescence) and TX (red fluorescence) channels. 

To quantify the levels of fluorescence indicative of live/dead cell labelling under these various conditions, images were opened on Zeiss ZEN 3.1 blue edition software (Carl Zeiss Microscopy GmbH, Munich, Germany) and the metatarsal area was manually traced. After tracing the area, the average fluorescence was obtained automatically by the software for both the FITC (live cells) and Texas Red (dead cells) channels. Graphs were then displayed by average fluorescence per µm^2^ to allow for direct comparison between metatarsi with different lengths.

### 4.8. Confocal Imaging

After 14 days in culture, metatarsi were collected in bijoux tubes and fixed in 10% neutral buffered formalin (NBF) for at least 24 h prior to clearing to facilitate 3D imaging of the entire immunochemically-labelled sample by confocal microscopy. Immunolabelling and clearing were developed based on methods published by Marr et al. [65]. Accordingly, metatarsi were processed according to Visikol^TM^ guidelines, and all steps were performed with orbital agitation at 60 RPM. Metatarsi were taken from the fixative and washed with TBS, then permeabilized using three graded methanol dilutions in TBS or dH_2_O (50% (*v*/*v* TBS), 80% (*v*/*v* dH_2_O), and 100% methanol) for a period of 1 h at 4 °C. Thereafter, samples were washed with detergent (0.2% Triton X-100) (20% *v*/*v*) DMSO:methanol, 80% methanol:dH_2_O, 50% methanol:TBS, 100% TBS and TBS:0.2% Triton) for 3.5h at 4 °C, followed by pre-block penetration with TBS, Triton X-100, glycine and DMSO (Sigma-Aldrich, USA) for 1 h at RT. This was followed by blocking of samples in 6% donkey and 6% goat serum (in pre-block penetration solution) for 3 h at 37 °C, and then incubation in anti-rabbit PCNA (1:50, Abcam, UK) primary antibody in TBS–Tween-20 (0.2% *v*/*v* ), 6% donkey serum, 6% goat serum and 5% DMSO at 37 °C overnight. Samples were then washed three times in TBS–Tween-20 (0.2% *v*/*v* ) for 1 h and then incubated in goat anti-rabbit Alexa Fluor 594 (1:500, Abcam, UK) secondary antibody in TBS–Tween-20 (0.2% *v*/*v*) at 37 °C for 8 h. Incubation in secondary antibody was followed by another series of washes in TBS–Tween-20 (0.2% *v*/*v*) and the sample was finally incubated in 4′,6-diamidino-2-phenylindole (DAPI, dilution 1:2000, Sigma-Aldrich, Germany) overnight before a final set of washes at 37 °C, and dehydration in graded methanol dilutions. After immunolabelling, samples were cleared in HISTO-1^TM^ and HISTO-2^TM^ solutions (proprietor supplied) using the Visikol^®^ kit and following the manufacturer’s instructions. Briefly, metatarsi were immersed in HISTO-1^TM^ solution for 1 h, followed by immersion in HISTO-2^TM^ solution for a minimum of 2 h. Samples were stored in HISTO-2^TM^ prior to imaging.

Samples were transferred to a confocal imaging plate with a glass bottom in HISTO-2^TM^ solution for imaging. Metatarsi were imaged on a Leica SP8 confocal microscope with a motorised stage (Leica Biosystems, Nussloch, Germany). Images were acquired with a 20x objective; pinhole size was set to 1 airy unit, line average was set to 48, and electronic zoom was set to 1.70. Sequential scans of samples (approx. 228 µm × 228 µm × 241 µm) were captured using lasers emitting light at 405 (blue channel; DAPI) and 561 (red channel; Alexa Fluor 594) nm to detect fluorescence with a low laser power (<10%), and 8000 Hz scanning speed. 3D analysis and visualisation were performed using Leica LAS-X v3.5.5 software (Leica Biosystems, Nussloch, Germany). To quantify the fluorescence under control and hydrogel-treated conditions, Leica image files (.lif) files were opened on Volocity software v6.3.0 (Quorum Technologies, Puslinch, Canada) and the metatarsal volume was traced using an intensity threshold to find all blue fluorescence (DAPI channel), then rejecting all but very large objects. Red fluorescence (PCNA channel) was obtained automatically after running the protocol for DAPI by re-using the same traced volume. Average fluorescence per µm^3^ is displayed to allow direct comparison of cell density labelling between metatarsi of divergent lengths. 

### 4.9. RNA Isolation

At each specified endpoint, metatarsi were collected and pooled (2 bones) to allow for isolation of sufficient RNA (~15–20 ng/µL); between 2–6 replicate pooled samples were analysed per condition (representative of *n* = 4–12 single bones). Accordingly, metatarsi were placed directly into a tube containing 200 µL of RNA later^TM^ stabilization solution (Invitrogen, Waltham, MA, USA) and stored at −80 °C until isolation. For RNA isolation, thawed metatarsals were transferred on ice to 1.5 mL RNAse-free tubes, containing 1 mL of QIAzol (Qiagen, Hilden, Germany) and manually homogenized by grinding with a plastic pestle. The homogenate was subsequently processed, and the RNA isolated using a RNeasy plus micro kit (Qiagen, Hilden, Germany), involving genomic DNA exclusion, chloroform incubation, RNA clean-up, membrane washing and mRNA elution (RNAse-free water)**.**

### 4.10. Multiplex Real Time Polymerase Chain Reaction

Prior to multiplex RT-PCR, mRNA integrity and concentration were determined using RNA Nano chips on a 2100 Bioanalyzer Instrument (Agilent, Santa Clara, CA, USA) following the manufacturer’s instructions. Samples with an RNA integrity number (RIN) ≥ 7 and a concentration between 15–20 ng/µL were considered of suitable quantity and purity for analysis. Primers were designed based upon FASTA sequences and accession numbers obtained from the NCBI website for: *Vegf*, *Sox9*, *Runx2*, *Alpl*, *Ihh, Tbsp4*, *Il6*, *Col10*, *Mmp13*, *Pthrp, Acan*, *Col2* and *Akt* and housekeeping genes *Actb* and *Gapdh*. Genes were selected as representative of early (*Runx2*, *Acan*, *Col2*, *Ihh*, *Pthrp*, *Sox9)* and late EO (*Mmp13, Alpl, Vegf, Col10*) or as were markers of mTOR (*Akt)* and NF-ĸB (*Il6*) pathway activation; *Tsp4* was also included. 

GenomeLab GeXP Genetic Analysis System (Sciex, Beckman and Coulter, Framingham, MA, USA) primer design tool was used to obtain the forward and reverse primer sequences (Table 3; Sigma-Aldrich, St. Louis, MO, USA). Primers were solubilised in RNAse/DNAse-free water and stored at −20 °C. Reverse transcriptase and PCR reactions were performed using GeXP protocols. Briefly, a primer master mix was prepared in two steps: a reverse transcription reaction and a forward PCR reaction. In the reverse transcription reaction, the stock concentrations of all the reverse primers were 100 µM and the working concentration was 0.5 µM (500 nM); in the forward PCR reaction, the stock concentration of all the forward primers was 100 µM and the working concentration was 0.2 µM (200 nM). RT reaction parameters: 48 °C for 1 min, 42 °C for 1 h, 95 °C for 5 min and hold at 4 °C. PCR reaction parameters: 95 °C for 10 min, 94 °C for 30 s—55 °C for 30 s—70 °C for 1 min and repeat for 35 cycles then hold at 10 °C. Raw data was obtained using GeXP software and the mRNA levels for *Gapdh* and β-actin (*Actb*) were obtained and results analysed using GraphPad Prism v9 (GraphPad Software Inc., San Diego, CA, USA). Data were normalised for both *Gapdh* and β-actin (*Actb*) housekeeping genes automatically using the GenomeLab GeXP Genetic Analysis System (Sciex, Beckman and Coulter, Framingham, MA, USA).

### 4.11. Assessment of mTOR and NF-ĸB Activity in Culture Supernatant

To quantify the mTOR and NF-ĸB activity present in *ex vivo* cultures, enzyme-linked immunosorbent assays (ELISAs) were performed using medium conditioned by metatarsals. All metatarsal-conditioned supernatants were collected following 14 days and stored at −80 °C until usage. IL-6 ELISA (Sigma-Aldrich, St. Louis, MO, USA) was used according to the manufacturer’s instructions; IL-6 was selected as a readily accessible downstream read-out of NF-ĸB pathway activity. Assays were performed in duplicate for each sample using a Tecan Pro 200 plate reader (Tecan, Männedorf, Switzerland).

### 4.12. Flow Cytometry

Flow cytometric analysis was applied to metatarsi cultured for 5, 7 and 14 days under either control and hydrogel-treated conditions; entire metatarsi were digested with collagenase IV [66] and subsequently stained with propidium iodide (PI) before analysis using the methods described by Kim and Cecchini [67,68]. Metatarsals were collected (4 to 10 pooled/sample) for enzyme digestion, which involved transferral to FACS tubes containing 1 mL of fresh culture medium (see [17]) supplemented with 86.5 U/mL collagenase IV (Worthington Biochem, Lakewood, NJ, USA) and 10 µL/mL of DNAse A (Invitrogen, USA), prior to incubation at 37 °C/5% CO_2_ and mixing by inversion every 15 min for 4 h. These samples were then centrifuged for 5 min at 300× *g* and the cell supernatant washed with fresh medium and then filtered (70 µm cell strainer) before re-centrifugation. Each cell suspension aliquot was then fixed by resuspension in ice-cold 70% ethanol for 30 min before another round of centrifugation and subsequent washing in ice-cold PBS (Gibco, Waltham, MA, USA) containing 0.2% BSA (Sigma-Aldrich, USA). Tubes were stored in the dark at 4 °C until staining.

For PI staining, cells were centrifuged for 5 min at 300× *g* (without active braking) and 900 µL of PBS/0.2% BSA was taken from each tube and 500 µL of staining solution containing 10 µg PI (Thermofisher Scientific, Waltham, MA, USA), 5 µg bovine RNAse A (Abcam, USA), 20% triton X-100 and PBS/0.2% BSA was added to each tube, and dye was incubated for 1 h at RT protected from light. After another round of centrifugation and two washes in PBS/0.2% BSA, 900 µL of PBS/0.2% BSA was added, and tubes were finally kept at 4 °C, protected from light until the flow cytometry. Cells were processed using a BD FACSCanto II Flow Cytometer (BD Biosciences, USA) and appropriate gating was applied to the expected PI pattern (30,000 events) on FlowJo v10.7.0 software (BD, Franklin Lanes, NJ, USA).

### 4.13. Statistical Analysis

Analyses of pooled data were performed using a linear mixed effect model in R v3.6.1 (R Foundation for Statistical Computing, Auckland, New Zealand) and RStudio v1.2.5019 (RStudio Inc., Boston, MA, USA) using xlsx, lattice, latticeExtra, lme4, lmerTest and phia packages and graphs were generated on GraphPad Prism 9 (GraphPad Software Inc., San Diego, CA, USA). Data are presented in intervals that represent the mean and SD. According to the normality test distribution, two variable comparison t-tests or Mann–Whitney tests were used. For more than two variables, one-way ANOVA was used. *p* values lower than 0.05 were considered statistically significant throughout.

## 5. Conclusions

The studies described here revealed the sensitivity of the interaction between the mTOR pathway and mechanics, with an inhibitor (rapamycin) and an activator (leucine) of mTOR both effectively reversing the arrest in growth exerted by hydrogel-mediated quasi static loading. mTOR is described for the first time as being a pathway with a bidirectional feedback capacity in the control of endochondral growth, as it seems to compensate for inhibition in one its complexes (mTORC1) by maintaining the other (mTORC2, and *vice-versa*; Figure 5).

## Figures and Tables

**Figure 1 ijms-22-13220-f001:**
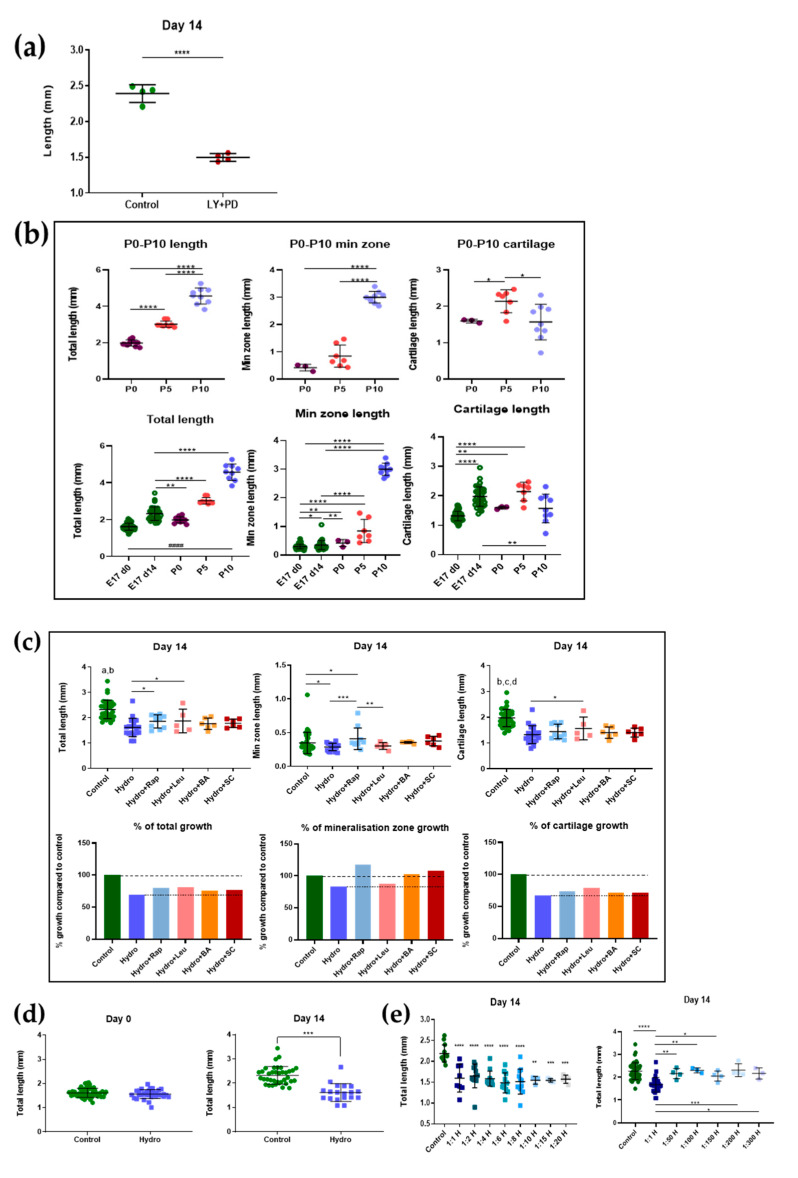
Effects of mTOR/NF-ĸB compounds and hydrogel on the growth of E17 metatarsi. (**a**)—differences in total length between control and growth inhibitors LY294002/PD98059. (**b**)—changes in total, mineralised zone, and cartilage zone lengths in mouse metatarsi at different embryonic and post-natal stages *in vivo* and comparison with changes during *ex vivo* maintenance culture. (**c**)—mTOR modulators, rapamycin, and leucine both somewhat overcome the hydrogel-mediated arrest in longitudinal growth. (**d**)—quasi-static loading almost completely arrests metatarsal longitudinal growth at days 7 and 14. (**e**)—hydrogel quasi-static loading shows dose-dependent restriction of longitudinal growth that rapidly switches above dilutions greater than 1:20. Data are presented in intervals that represent the mean and SD. Results were analysed with a linear mixed effect model and ANOVA test according to normality test distribution. Hydro/H—hydrogel; Rap—rapamycin; Leu—leucine; BA—betulinic acid; SC—SC-514. **** *p* < 0.0001, *** *p* < 0.001, ** *p* < 0.01, * *p* < 0.05, a—control vs. all conditions except hydrogel + leucine *p* < 0.001, b—control vs. hydrogel + leucine *p* < 0.05, c—control vs. hydrogel and hydrogel + rapamycin *p* < 0.0001, d—control vs. hydrogel + BA and hydrogel + SC *p* < 0.001.

**Figure 2 ijms-22-13220-f002:**
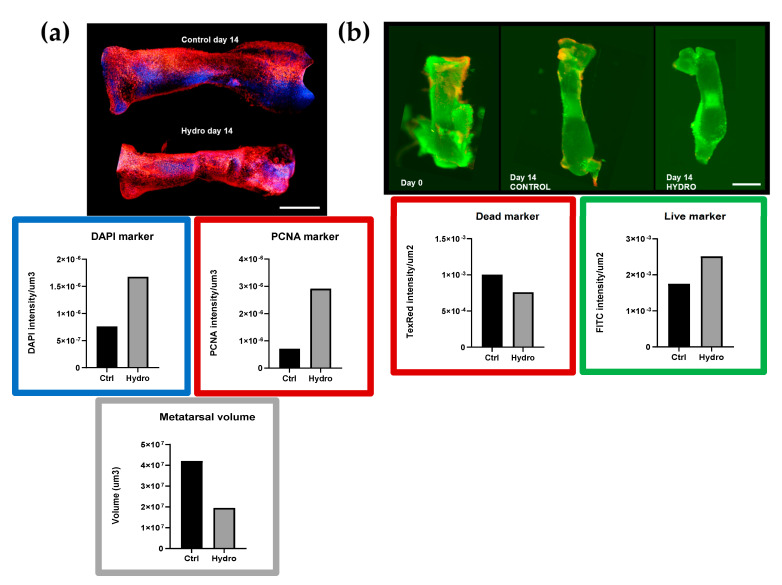
Metatarsi stained for PCNA and Live/Dead cell labelling. (**a**)—cultured metatarsi in hydrogel show high levels of PCNA staining at day 14. PCNA 3D confocal whole-mount imaging of metatarsi for proliferation assays, showing DAPI (blue box) and PCNA (red box) mean channel intensity per µm^3^ of volume. Hydrogel displays higher mean values than the control for both. (**b**)—metatarsi cultured in hydrogel label positively for viable cells at day 14. 2D fluorescence imaging of metatarsi for live/dead assay (viability). Differences in cell death between control and hydrogel using mean intensity of Texas Red channel per µm^2^ of area (red box) and live cells using mean intensity of FITC channel per µm^2^ of area (green box). Hydrogel displays less cell death and brighter live staining than controls. These data are a representation of observations made using 3D live/dead exclusion assay in >1 tarsals. For quantification, 1 sample per condition was analysed. Ctrl—control; Hydro—VitroGel3D hydrogel; Scale bar = 5 mm.

**Figure 3 ijms-22-13220-f003:**
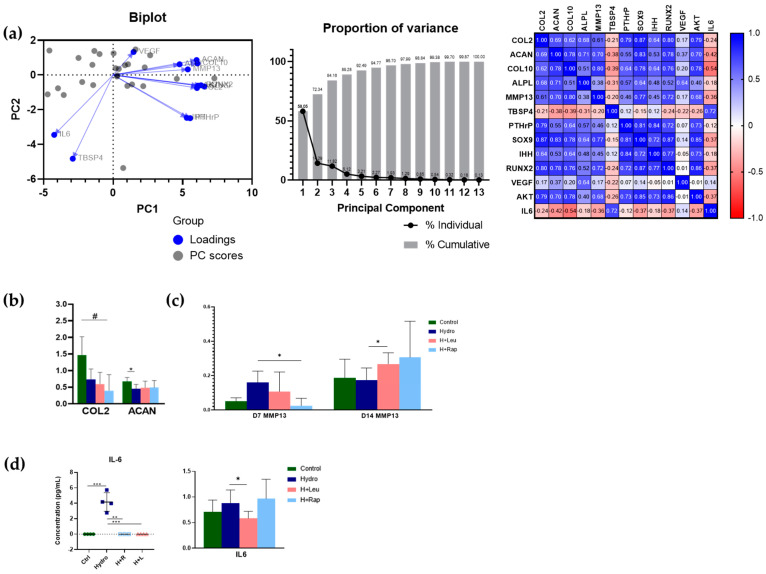
mRNA expression shows that hydrogel quasi-static loading interacts with the mTOR pathway. (**a**)—characterisation of multiplex RT-PCR results using principal component analysis, which shows that variance in dataset can be explained mostly by changes in *Col2* and *Acan* mRNA levels. Biplot showing principal component (PC) 1 and 2, *Col2* and *Acan*, respectively, along with their PC scores; this shows that most genes in the EO multiplex behave similarly, except for *Vegf*, *Il6* and *Tbsp4*. PCs 1 and 2 (*Col2* and *Acan*) together account for 72% of variance in the dataset, which increases to 84% with the addition of PC3 (*Col10*). Heatmap shows the 13 analysed genes with PC scores and how they correlate. (**b**)—day 7 *Col2* and *Acan* mRNA levels are downregulated in the hydrogel condition. (**c**)—*Mmp13* mRNA levels at days 7 and 14 show modulation by either leucine (day 7) or rapamycin (day 14). (**d**)—IL-6 protein levels determined by ELISA in culture supernatants (left). Marked increases in IL-6 levels were seen in media conditioned by metatarsals in hydrogel. *Il6* mRNA levels were downregulated by leucine when added to hydrogel (right). For mRNA, *n* = 2–6 samples per condition. Data are presented as mean and SD. Results were analysed with ANOVA and t-tests according to normality test distribution. IL-6 ELISA was analysed using Grubbs method of outlier calculation on GraphPad Prism. *** *p* < 0.001, ** *p* < 0.01, * *p* < 0.05, # *p* < 0.05 between control and all conditions. Ctrl—control; Hydro—hydrogel; Rap—rapamycin; Leu—leucine; H + R—hydrogel + rapamycin; H + L—hydrogel + leucine.

**Figure 4 ijms-22-13220-f004:**
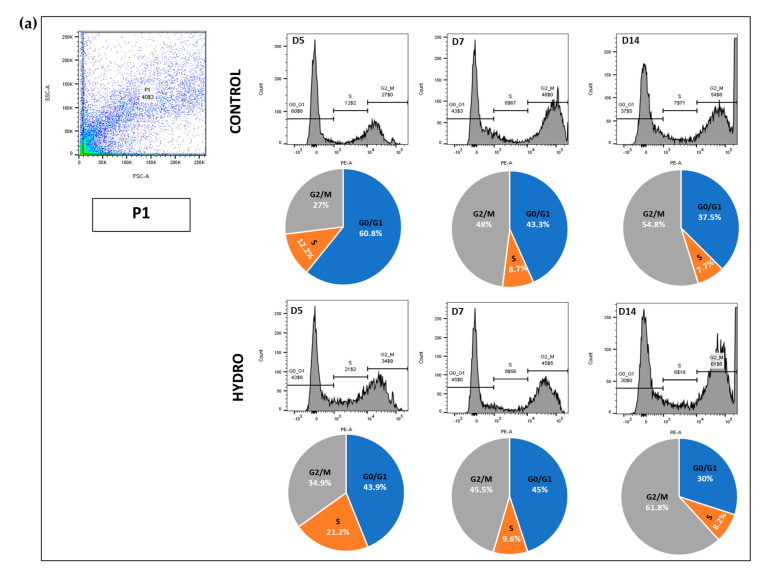
Representations of PI-labelled P1 and P2–P4 populations as scatter plots and d5, d7 and d14 as histograms. (**a**)—representation of the PI-labelled P1 population as a scatter plot and d5, d7 and d14 as histograms. Pie charts represent the proportion of the total number of cells in each cell cycle phase (G0/G1, S and G2/M) in cell populations isolated from metatarsals maintained for 5 (left), 7 (middle) and 14 (right) days in the absence (control) and presence (hydro) of hydrogel quasi-static loading conditions. (**b**)—representation of the PI-labelled P2–P4 populations pooled together as a scatter plot and d5, d7 and d14 pie charts. Pie charts represent the proportion of the total number of cells in each cell cycle phase (G0/G1, S and G2/M) in cell populations isolated from metatarsals maintained for 5 (left), 7 (middle) and 14 (right) days in the absence (control) and presence (hydro) of hydrogel quasi-static loading conditions. Data were analysed using BD FACSDIVA and FlowJo. Ctrl—control; Hydro—hydrogel; d5—day 5; d7—day 7; d14—day 14; SSC-A—side scatter area; FSC-A—forward scatter area; PE-A—phycoerythrin fluorescent dye area; PE-W—phycoerythrin fluorescent dye width.

**Figure 5 ijms-22-13220-f005:**
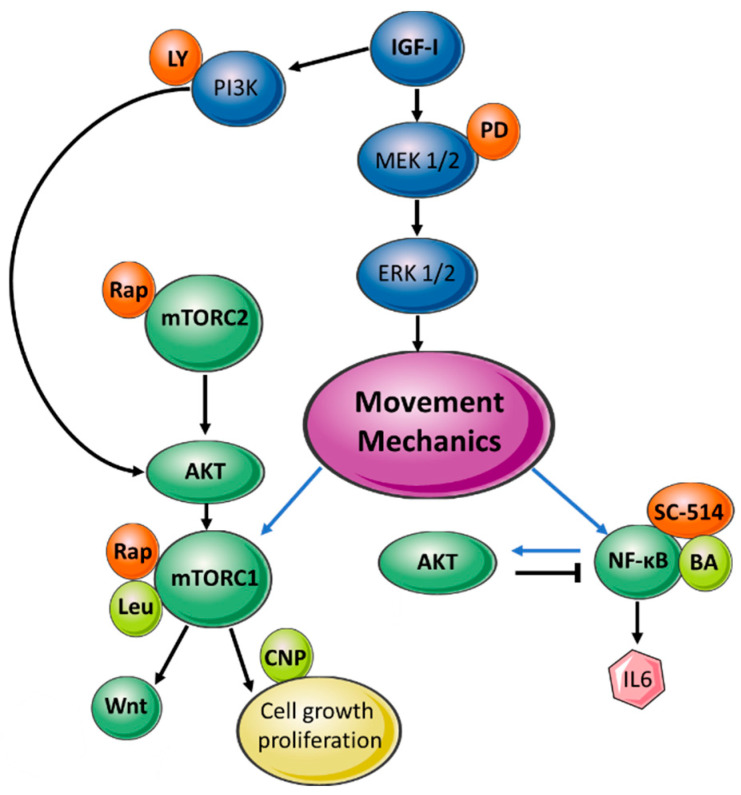
Summary of NF-ĸB/mTOR and interaction with mechanics. Dark green circles—pathway factors; yellow circles—external factors; light green circles—activator compound; orange circles—inhibitor compound; blue circles—IGF-I pathway; purple circles—movement/loading; pink hexagon—readout markers; black arrows—interactions between mTOR, NF-ĸB and AKT; blue arrows—effects of quasi-static loading. BA—betulinic acid; Rap—rapamycin; Leu—leucine; PD—PD98059; LY—LY294002; CNP—C-type natriuretic peptide.

**Table 1 ijms-22-13220-t001:** Selected modulators of NF-ĸB and mTOR pathways.

Compounds	Function	Concentration
Betulinic acid	NF-ĸB activator	2.5 µM
SC-514	NF-ĸB inhibitor	20 µM
Leucine	mTOR activator	10 mM
Rapamycin	mTOR inhibitor	100 nM
Vitrogel 3D™ (Hydrogel)	Quasi-static loading	1:1 dilution

**Table 2 ijms-22-13220-t002:** Summary of the cell cycle analysis of control and hydrogel conditions at days 5, 7 and 14. Hydro—hydrogel; P—population.

	Control vs. Hydro (%)			
Day/Cycle Phase	P1 (All Cells)	P2 (Pre-Hypertrophic)	P3 (Proliferative)	P4 (Hypertrophic)
Day 5/G0 and G1	60.8 vs. 43.9	0	48.9 vs. 38.3	0
5/S	12.2 vs. 21.2	15.8 vs. 14.6	51.1 vs. 61.7	0 vs. 9.5
5/G2 and M	27 vs. 34.9	84.2 vs. 85.4	0	100 vs. 90.5
Day 7/G0 and G1	43.3 vs. 45	0	55.3 vs. 45.7	0
5/S	8.7 vs. 9.6	2.5 vs. 4.7	44.7 vs. 54.3	0
5/G2 and M	48 vs. 45.5	97.5 vs. 95.3	0	100
Day 14/G0 and G1	37.5 vs. 30	0	54.9 vs. 47.1	0
5/S	7.7 vs. 8.2	2.3 vs. 2.9	45.1 vs. 52.9	5.5 vs. 3.6
5/G2 and M	54.8 vs. 61.8	97.7 vs. 97.1	0	94.5 vs. 96.4

**Table 3 ijms-22-13220-t003:** List of primers used for multiplex RT-PCR.

Gene		Sequence (5′-3′)
** *Vegf* **	F	AGGTGACACTATAGAATAGCTGTGTGTGTGAGTGGCTT
	R	GTACGACTCACTATAGGGACTCTTTTCTCTGCCTCCGTG
** *Sox9* **	F	AGGTGACACTATAGAATAAGGAAGCTGGCAGACCAGTA
	R	GTACGACTCACTATAGGGACGTTCTTCACCGACTTCCTC
** *Runx2* **	F	AGGTGACACTATAGAATAACAGTCCCAACTTCCTGTGC
	R	GTACGACTCACTATAGGGATAGTTCTCATCATTCCCGGC
** *Alpl* **	F	AGGTGACACTATAGAATACACTCAAGGGAGAGGTCCAG
	R	GTACGACTCACTATAGGGACCCAAGAGAGAAACCTGCTG
** *Ihh* **	F	AGGTGACACTATAGAATACCGAACCTTCATCTTGGTGT
	R	GTACGACTCACTATAGGGACCCCGAGAAACATTGGAGTA
** *Tbsp4* **	F	AGGTGACACTATAGAATATTCAGTCCCCAACTCCAAAC
	R	GTACGACTCACTATAGGGACGTTTCCCGTGTAACCATCT
** *Il6* **	F	AGGTGACACTATAGAATAAGTTGCCTTCTTGGGACTGA
	R	GTACGACTCACTATAGGGAAGCCTCCGACTTGTGAAGTG
** *Col10* **	F	AGGTGACACTATAGAATAGCAATTGCAGAAAGTCCACA
	R	GTACGACTCACTATAGGGACTCGATTGAAAGGCACACAA
** *Mmp13* **	F	AGGTGACACTATAGAATACCAGAACTTCCCAACCATGT
	R	GTACGACTCACTATAGGGAGTCTTCCCCGTGTTCTCAAA
** *Pthrp* **	F	AGGTGACACTATAGAATATTCCTGCTCAGCTACTCCGT
	R	GTACGACTCACTATAGGGAGGTAGCTCTGATTTCGGCTG
** *Acan* **	F	AGGTGACACTATAGAATAAGGACTGAAATCAGCGGAGA
	R	GTACGACTCACTATAGGGATGTCTCTGTAGGGTACCGGG
** *Col2* **	F	AGGTGACACTATAGAATAACACTGGTAAGTGGGGCAAG
	R	GTACGACTCACTATAGGGATCGCAATGGATTGTGTTGTT
** *Akt* **	F	AGGTGACACTATAGAATAGCAGTGGACCACAGTCATTG
	R	GTACGACTCACTATAGGGACATCGTCTCTTCTTCCTGCC
** *Gapdh* **	F	AGGTGACACTATAGAATAGGGTGTGAACCACGAGAAAT
	R	GTACGACTCACTATAGGGAACTGTGGTCATGAGCCCTTC
** *Actb* **	F	AGGTGACACTATAGAATAGTACCACCATGTACCCAGGC
	R	GTACGACTCACTATAGGGAGTACTTGCGCTCAGGAGGAG

## Data Availability

Not applicable.

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
