# Peer review of "Restraint upon Embryonic Metatarsal *Ex Vivo* Growth by Hydrogel Reveals Interaction between Quasi-Static Load and the mTOR Pathway"

_ijms, 2021, doi:10.3390/ijms222413220_

Round 1
Reviewer 1 Report
WHAT IS NEW HERE? PLEASE PUT WHAT IS NEW IN TITLE, ABSTRACT AND IN CONCLUSION! IN CONCLUSIONS, WRITE ONLY THE NEW!
Author Response
We appreciate the comments made regarding the need to better emphasise the novelty. The novelty of this study is the connection between mTOR and mechanics, namely the loading regime used with a commercially available hydrogel (quasi-static loading). This is clearly stated in the title and abstract. Changes have, however, been made to the conclusions. We hope this has made the innovation and novelty provided by our studies clearer.
Reviewer 2 Report
This manuscript describes the use of an adapted ex vivo murine metatarsi assay based on a loading regime that uses a commercial hydrogel, to explore endochondral ossification. The results obtained showed the sensitivity of the interaction between the mTOR pathway and mechanics. The major problem with this manuscript is that the results and discussion sections are difficult to understand. The authors should re-write these sections in order to show clearly the objectives of the study and the importance of the results obtained. The authors should remove part of the figures and graphics from the manuscript and include them in the supporting information file. The importance of using the hydrogel is not clear.
Author Response
We appreciate the feedback on the manuscript, specifically on need to streamline the Results/Discussion sections. Changes were made to include the importance of the results obtained, namely the novelty of this study and the importance of the hydrogel in this context. The Results/Discussion sections have also been made shorter and clearer. They now, we feel, better summarise the findings in a more straightforward way, and we have also made changes in the sub-headings and modified the short sentences at the end of each sub-section to better recapitulate the outcomes.
Reviewer 3 Report
This is a brilliant study on the assessment on endochondral ossification in embryonic metatarsal bones under different biological and mechanical cues. The study is well structured, the methodology is strong and the results are intriguing and well discussed. I strongly believe that this paper will enrich the audience of IJMS.
Only some minor comments:
- Line 39: The extended form of BMP9 and FGF should be mentioned, with the abbreviated one in brackets. Same for, PTHrP, IHH, ALP, VEGF, MMP and so on.
- Line 41: Probably authors mean "more distant".
- The Introduction section is too long and should be shortened. Lines 52-54 (from "Zebrafish..." to "...setting"), lines 64-69 (from "NFkB..." to "...activity). lines 71-79 (from "mTOR") should be removed. Details about agonists and inhibitors may be reconsidered in the Methods/Discussion sections where applicable.
- Authors should reconsider the significant use of italic throughout the manuscript.
Author Response
We appreciate the feedback on the manuscript. All of the suggested changes have been made: i) changes were made in the abbreviations used ii) the paragraphs pinpointed in the Introduction have accordingly been moved to the methods section. We are grateful for these suggested improvements. Italic is now only used for Latin words or mouse genes.